# A bacterial cell factory converting glucose into *scyllo*-inositol, a therapeutic agent for Alzheimer's disease

Christophe Michon[1,3], Choong-Min Kang[2], Sophia Karpenko[1,4,5,6], Kosei Tanaka[1], Shu Ishikawa[1] & Ken-ichi Yoshida [1✉]

A rare stereoisomer of inositol, *scyllo*-inositol, is a therapeutic agent that has shown potential efficacy in preventing Alzheimer's disease. *Mycobacterium tuberculosis ino1* encoding *myo*-inositol-1-phosphate (MI1P) synthase (MI1PS) was introduced into *Bacillus subtilis* to convert glucose-6-phosphate (G6P) into MI1P. We found that inactivation of *pbuE* elevated intracellular concentrations of $NAD^+ \cdot NADH$ as an essential cofactor of MI1PS and was required to activate MI1PS. MI1P thus produced was dephosphorylated into *myo*-inositol by an intrinsic inositol monophosphatase, YktC, which was subsequently isomerized into *scyllo*-inositol via a previously established artificial pathway involving two inositol dehydrogenases, IolG and IolW. In addition, both *glcP* and *glcK* were overexpressed to feed more G6P and accelerate *scyllo*-inositol production. Consequently, a *B. subtilis* cell factory was demonstrated to produce $2 \text{ g L}^{-1}$ *scyllo*-inositol from $20 \text{ g L}^{-1}$ glucose. This cell factory provides an inexpensive way to produce *scyllo*-inositol, which will help us to challenge the growing problem of Alzheimer's disease in our aging society.

[1] Department of Science, Technology and Innovation, Kobe University, Kobe 657 8501, Japan. [2] Department of Biological Sciences, California State University, Stanislaus, Turlock, CA 95382, USA. [3] Present address: CHROMagar, 4 Place du 18 Juin 1940, 75006 Paris, France. [4] Present address: Sorbonne Universités, UPMC Univ. Paris 06, UMR 8237, Laboratoire Jean Perrin, F-75005 Paris, France. [5] Present address: CNRS UMR 8237, Laboratoire Jean Perrin, F-75005 Paris, France. [6] Present address: Paris Sciences & Lettres, 60 rue Mazarine, F-75006 Paris, France. ✉email: kenyoshi@kobe-u.ac.jp

Dementia, most commonly caused by Alzheimer's disease, reached a prevalence of 50 million people worldwide in 2018; this number is expected to increase to 152 million by 2050, with the biggest increase occurring in developing countries[1]. Alzheimer's disease involves the aggregation of amyloid β-proteins, engendering apoptosis of neurons and loss of cognitive function[1]. Currently there is no cure for Alzheimer's disease, but numerous attempts have been made to develop molecules capable of targeting these aggregations of amyloid β-proteins. Among these molecules, a rare stereoisomer of inositol, scyllo-inositol, has been shown to be promising. When taken orally, scyllo-inositol is able to reach the brain, and prevents amyloid β-proteins forming toxic amyloid fibrils and polymers[2]. scyllo-Inositol administered at 250 mg per day has been shown to have an acceptable level of safety. Patients given this dose had higher scyllo-inositol concentrations and fewer amyloid plaques in their cerebrospinal fluid. Slight but significant increases in brain ventricular volume were observed, but other markers of Alzheimer's disease were unchanged. The small sample size of 250 mg per day did not provide evidence to support or refute any benefits associated with scyllo-inositol[3]. Following these results, however, there are now plans to test scyllo-inositol in phase III clinical trial[4]. Other studies, where scyllo-inositol treatment has been combined with other treatments, such as antibodies against amyloid β-proteins[5], or using a guanidine-appended scyllo-inositol derivative[6] have shown great potential of scyllo-inositol or its derivatives to treat Alzheimer's disease in animal models. However, such trials and the possible development of treatments require a large quantity of scyllo-inositol, which rarely occurs in nature. Currently, the starting compound for the commercial production of scyllo-inositol, phytic acid, is an inexpensive derivative of another stereoisomer of myo-inositol, and is extracted from fruits, beans, grains, and nuts[7]. Phytic acid is first chemically converted into myo-inositol and then further transformed into scyllo-inositol by an expensive enzymatic/chemical conversion[8,9].

B. subtilis possesses the iolABCDEFGHIJ operon, which encodes the enzymes responsible for the metabolism of inositol. This metabolism includes IolG, an inositol dehydrogenase that couples the reduction of NAD$^+$ with the oxidation of myo-inositol to give scyllo-inosose (Fig. 1). Another gene, iolX, encodes a different inositol dehydrogenase and enables the catabolism of scyllo-inositol through the oxidation of scyllo-inositol to scyllo-inosose coupled with the reduction of NAD$^+$. scyllo-Inosose is further degraded, enabling it to enter glycolysis and the citric acid cycle following successive reactions catalyzed by the enzymes IolE, IolD, IolB, IolC, IolJ, and IolA. In addition, both IolF and IolT function in transporting inositol into the cell, while IolR

and IolQ act as repressors regulating transcription of the iol genes[10–14]. An additional inositol dehydrogenase, IolW, is a key enzyme that enables B. subtilis to produce scyllo-inositol by specifically reducing scyllo-inosose into scyllo-inositol, coupled with the oxidation of NADPH (Fig. 1)[12]. In the last decade, B. subtilis has been genetically engineered to form a "cell factory" capable of performing the bioconversion of myo-inositol into rare inositol stereoisomers[15–19]. One of the cell factories recently created was capable of producing 27.6 g L$^{-1}$ of scyllo-inositol from 50 g L$^{-1}$ of myo-inositol within 48 h of culturing[15]. In this case, iolG, iolW, and iolT were simultaneously overexpressed in a strain that lacked all of the other iol genes. In addition, a heterologous nicotinamide nucleotide transhydrogenase capable of regenerating NADPH was introduced, since IolW requires NADPH to reduce scyllo-inosose into scyllo-inositol[14]. The bioconversion is efficient enough to enable a high productivity, but the scyllo-inositol produced can never be cheaper than the starting material, myo-inositol.

In natural biological systems, the biosynthesis of myo-inositol from glucose is widely conserved, especially in eukaryotes, due to the importance of myo-inositol as a moiety of the phospholipid phosphatidylinositol, which is found in the plasma membrane, and also because myo-inositol acts as a second messenger in the cell system[20]. myo-Inositol biosynthesis includes three steps: the phosphorylation of glucose into glucose-6-phosphate (G6P), commonly seen in many organisms; the conversion of glucose-6-phosphate (G6P) into myo-inositol 1-phosphate (MI1P) by MI1P synthase (MI1PS) (EC:3.1.3.25) encoded by ino1; and cleaving-off of a phosphate from MI1P to form myo-inositol by inositol monophosphatase (IMP) (Fig. 1). The key enzyme MI1PS is also found in some archaea and bacteria, including Mycobacterium tuberculosis, which, like B. subtilis, is a Gram-positive bacteria and possesses the myo-inositol biosynthesis pathway involving a functional ino1[21]. Within the genome of B. subtilis, there is no gene likely to encode MI1PS, but there is an intrinsic yktC gene encoding functional inositol monophosphatase[22]. Therefore, we speculated that simply expressing M. tuberculosis ino1 in B. subtilis may enable myo-inositol biosynthesis, and this myo-inositol can be introduced to a previously established cell factory platform that converts myo-inositol into scyllo-inositol[14], enabling the production of scyllo-inositol from glucose in a single bacterial cell factory (Fig. 1).

In this study, we demonstrate a B. subtilis cell factory to produce 2 g L$^{-1}$ scyllo-inositol from 20 g L$^{-1}$ glucose. This cell factory provides an inexpensive way to produce scyllo-inositol.

## Results

**Introduction of MI1PS into B. subtilis.** M. tuberculosis has efficient myo-inositol biosynthesis involving ino1 encoding MI1PS[23,24]. In order to introduce functional M. tuberculosis ino1, its codon usage was optimized for expression in B. subtilis (Supplementary Fig. 1). The modified ino1 was cloned into the amyE locus on the chromosome of B. subtilis strain 168 to be expressed as a C-terminal His-tag fusion under the strong and constitutive rpsO promoter to give strain TK002 [amyE::(PrpsO-ino1Mt-His$_6$ cat)] (Supplementary Table 1).

The ino1 transcript was accumulated during the culture process (Fig. 2a). In addition, MI1PS protein as a His-tag fusion was detected by Western blot analysis (Fig. 2b). These results suggested that M. tuberculosis ino1 was expressed, producing MI1PS as a protein in B. subtilis. However, MI1PS activity in TK002 could not be detected even in the presence of 10 mM NAD$^+$ (Fig. 3a). In a previous study, the production of heterologous MI1PS was problematic due to its misfolding during the translation process[25]. We therefore tried to express

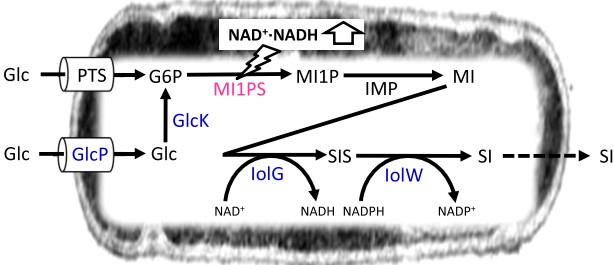

**Fig. 1 Strategy for the production of myo-inositol and scyllo-inositol from glucose in B. subtilis.** Reactions for the production of myo-inositol and scyllo-inositol from glucose in B. subtilis are represented schematically, where compound and enzyme names are abbreviated as myo-inositol (MI), scyllo-inositol (SI), scyllo-inosose (SIS), and described in the main text. The enzyme shown in magenta was introduced heterogeneously, while those in blue overexpressed.

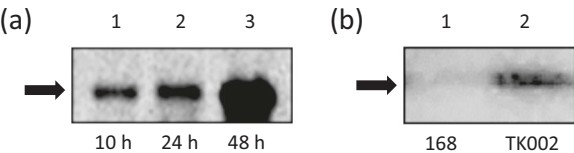

**Fig. 2 Expression of *ino1* for MI1PS in *B. subtilis* strain TK002 [*amyE*::(P*rpsO-ino1Mt-His$_6$ cat*)]. a** Transcription of *ino1* in *B. subtilis*. The *ino1* transcript was detected at the expected size (1.1 kb) by Northern blot analysis and accumulated as the cells grew after culturing for 10, 24, and 48 h (lanes 1, 2, and 3, respectively; each lane contained 30 µg of the RNA extract.). The arrowhead indicates the position of bands for the transcript. Similar results were obtained for three independently repeated experiments, and a representative result is shown. (The original gel image is supplied as Supplementary Fig. 4). **b** Production of MI1PS as a C-terminal His-tag fusion in *B. subtilis*. The protein extracts (50 µg per lane) were subjected to SDS-PAGE followed by the Western blot analysis with the anti-His-tag antibody. The MI1PS protein was detected at the expected size (41 kDa) in strain TK002 (lane 2) but not in strain 168 (lane 1) after culturing for 24 h. The arrowhead indicates the position of the band for the protein. Similar results were obtained for three independently repeated experiments, and a representative result is shown. (The original gel image is supplied as Supplementary Fig. 5).

*ino1* under another, weaker promoter, P*ybfK*, in an additional newly constructed strain, TK003 (Supplementary Table 1), anticipating that the reduced transcription level might assist in the correct translation and folding process. However, no MI1PS activity was seen in this case, either (Fig. 3a). These results suggested that *M. tuberculosis* MI1PS produced in *B. subtilis* might inevitably be misfolded or be inactive for other unknown reasons.

**Inactivation of *pbuE* restored MI1PS activity in *B. subtilis*.** It is known that the MI1PS enzyme requires NAD$^+$·NADH as an essential cofactor to become functional, and that NAD$^+$·NADH has to be properly located within the Rossmann-fold domain of the enzyme during its translation[26,27]. We happened to discover that *B. subtilis* strain YDHLd, which had lost *pbuE*, a gene that encodes a purine base/nucleoside efflux pump[28] [*pbuE*::pMutin2(*erm*)] (Supplementary Table 1), showed significantly elevated intracellular levels of NAD$^+$·NADH compared with its parental strain 168 (Fig. 3b); the concentration was up to 20 mM, which was almost twice more than that in strain 168 cells.

Strain KS001, which had lost all of its genes for *myo*-inositol catabolism (Δ*iolABCDEFGHIJ* Δ*iolX* Δ*iolR*), was transformed using DNA from YDHLd [*pbuE*::pMutin2(*erm*)] to give MC010 [*pbuE*::pMutin2(*erm*) Δ*iolABCDEFGHIJ* Δ*iolX* Δ*iolR*] (Supplementary Table 1). In addition, the *M. tuberculosis ino1* cassette of TK002 [*amyE*::(P*rpsO-ino1Mt-His$_6$ cat*)] was introduced into KS001 and MC010 to give MC001 [*amyE*::(P*rpsO-ino1Mt-His$_6$ cat*) Δ*iolABCDEFGHIJ* Δ*iolX* Δ*iolR*] and MC011 [*amyE*::(P*rpsO-ino1Mt-His$_6$ cat*) *pbuE*::pMutin2(*erm*) Δ*iolABCDEFGHIJ* Δ*iolX* Δ*iolR*], respectively. Of the strains constructed as described above, only MC011 exhibited significant MI1PS activity in its cell extract (Fig. 3a).

These results indicated that the inactivation of *pbuE* increased the intracellular levels of NAD$^+$·NADH, and it is likely that these elevated levels of NAD$^+$·NADH in vivo were able to restore or stabilize *M. tuberculosis* MI1PS activity in *B. subtilis*.

**SI production in *B. subtilis*.** As described above, in *B. subtilis* MC011, which lacked both functional *pbuE* and *myo*-inositol catabolism, the artificially introduced *M. tuberculosis ino1* was "restored" to produce an active enzyme. This could enable the

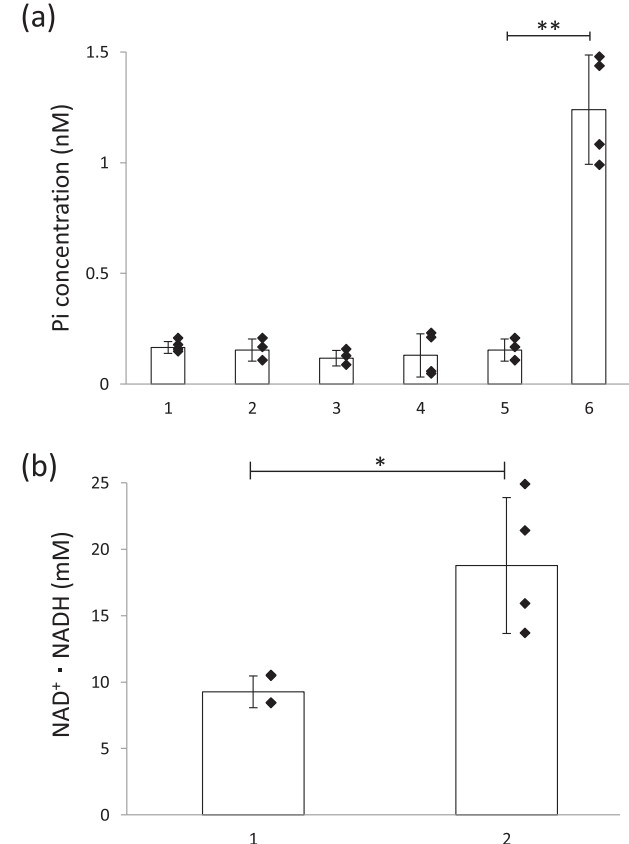

**Fig. 3 MI1PS activity and NAD$^+$·NADH levels in different strains of *B. subtilis*. a** MI1PS activity in strains 168 (parental strain, column 1), TK002 [*amyE*::(P*rpsO-ino1Mt-His$_6$ cat*), column 2], TK003 [*amyE*::(P*ybfK-ino1Mt-His$_6$ cat*), column 3], MC010 [*pbuE*::pMutin2(*erm*) Δ*iolABCDEFGHIJ* Δ*iolX* Δ*iolR*, column 4], MC001 [*amyE*::(P*rpsO-ino1Mt-His$_6$ cat*) Δ*iolABCDEFGHIJ* Δ*iolX* Δ*iolR*, column 5], and MC011 [*amyE*::(P*rpsO-ino1Mt-His$_6$ cat*) *pbuE*::pMutin2(*erm*) Δ*iolABCDEFGHIJ* Δ*iolX* Δ*iolR*, column 6]. MI1PS activity in cells cultured for 24 h was measured as described in the Methods[45]. The activity was expressed as the concentration of inorganic phosphate (Pi) present in the assay mixture. The values are means ± SEM from at least three independently repeated experiments (Supplementary Data 1; the individual data-points are plotted as closed diamonds.) \*\*Statistical significance was calculated using the Mann–Whitney U test with Prism (GraphPad software) based on the difference between MC001 and MC011 ($p < 0.01$). **b** NAD + ·NADH levels in strains 168 (parental strain, column 1) and YDHLd [*pbuE*::pMutin2(*erm*), column 2]. The total concentration of NAD$^+$ and NADH in cells after culturing for 24 h was measured as described in the Online Methods. The values are means ± SEM from at least three independent experiments (Supplementary Data 2; the individual data-points are plotted as closed diamonds.). \*Statistical significance was calculated using the Mann–Whitney U test with Prism (GraphPad software) ($p < 0.05$).

conversion of glucose into *myo*-inositol, since its substrate G6P is naturally supplied from glucose and also because its product MI1P is dephosphorylated by the intrinsic and constitutive YktC to form *myo*-inositol[22]. Accordingly, in order to enable the production of *scyllo*-inositol from glucose, we next introduced a previously established artificial pathway to convert *myo*-inositol into *scyllo*-inositol involving two inositol dehydrogenases, IolG and IolW[15]. To achieve this, strain MC022 [*amyE*::(P*rpsO-iolG-iolW-iolT kan*) *pbuE*::(P*rpsO-ino1Mt-His$_6$ cat*) Δ*iolABCDEF* Δ*iolHIJ* Δ*iolX* Δ*iolR*] was newly constructed, with *pbuE* disrupted by the insertion of an *ino1* gene cassette on the KU302 background to couple *myo*-inositol production from glucose with the

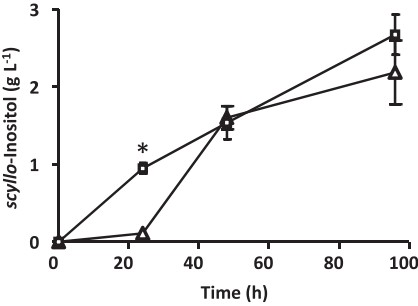

**Fig. 4 scyllo-Inositol production in B.** subtilis strains MC022 [ΔiolABCDEF ΔiolHIJ ΔiolX ΔiolR amyE::(PrpsO-iolG-iolW-iolT kan) pbuE::(PrpsO-ino1Mt-His₆ cat), triangles] and MC031 [ΔiolABCDEF ΔiolHIJ ΔiolX ΔiolR amyE:: (PrpsO-iolG-iolW-iolT kan) pbuE::(PrpsO-ino1Mt-His₆ cat) epr::(PrpsO-glcP-glcK ble), squares]. The concentration of scyllo-inositol in the culture media was measured over the course of the culture time. The values are means ± SEM from four independent experiments (Supplementary Data 3). *Statistical significance was calculated using the Mann–Whitney U test with Prism (GraphPad software) ($p < 0.05$).

conversion of *myo*-inositol into *scyllo*-inositol (Supplementary Table 1). MC022 was grown in Soytone medium containing 20 g L$^{-1}$ glucose, and *scyllo*-inositol increasingly appeared in the culture medium as the growth time was extended. The amount of *scyllo*-inositol produced was $0.11 \pm 0.15$ g L$^{-1}$, $1.60 \pm 0.33$ g L$^{-1}$, and $2.19 \pm 0.92$ g L$^{-1}$ after being cultured for 24, 48, and 96 h, respectively (triangles, Fig. 4). These results demonstrated that we successfully created the first bacterial cell factory for the production of *scyllo*-inositol from glucose.

In *B. subtilis*, it is known that the phosphoenolpyruvate-dependent phosphotransferase system (PTS) functions as the major mechanism for the uptake of glucose[29,30]. Therefore, we introduced an additional mechanism for the uptake of glucose involving the GlcP glucose transporter[29] and GlcK glucose kinase[30], anticipating that this could feed more G6P to be converted into *myo*-inositol. Thus, strain MC031 [epr::(PrpsO-glcP-glcK ble) amyE::(PrpsO-iolG-iolW-iolT kan) pbuE::(PrpsO-ino1Mt-His₆ cat) ΔiolABCDEF Δiol-HIJ ΔiolX ΔiolR] was constructed, in which both *glcP* and *glcK* were overexpressed in a single artificial operon [epr::(PrpsO-glcP-glcK ble)] on an MC022 background (Supplementary Table 1). The overexpression of *glcP* and *glcK* did not significantly affect the growth (Supplementary Fig. 2). MC031 was grown and produced $0.95 \pm 0.12$ g L$^{-1}$ and $2.68 \pm 0.45$ g L$^{-1}$ *scyllo*-inositol after being cultured for 24 and 96 h, respectively (Fig. 4). These results indicated that in MC031 the simultaneous overexpression of *glcK* and *glcP* could accelerate the rate of *scyllo*-inositol production by feeding more G6P to the earlier stage compared with MC022, although this failed to significantly increase the final concentration.

## Discussion

In the present study, we demonstrated the production of *scyllo*-inositol from glucose using a new *B. subtilis* cell factory, by combining two conversion steps: one step from glucose to *myo*-inositol and the second step from *myo*-inositol to *scyllo*-inositol. To develop the former step and convert glucose into *myo*-inositol in *B. subtilis*, we introduced *M. tuberculosis ino1*, encoding MI1PS as the key enzyme. MI1PS has been reported to be a rate-limiting enzyme for the synthesis of *myo*-inositol in natural biological systems[31]. The MI1PS enzyme was first described in parsley leaves, rat testes, and *Saccharomyces cerevisiae*[32–35]. MI1PS requires NAD$^+$·NADH as an essential cofactor, since the reaction catalyzed by this enzyme involves the oxidation of G6P coupled with the reduction of NAD$^+$ to form an intermediate, *myo*-2-

inosose-1-phosphate, which is reduced to the product MI1P through being coupled with the oxidation of the previously formed NADH[26]. The cofactor NAD$^+$·NADH has been shown to be localized in the Rossmann-fold domain of MI1PS, a domain which is conserved throughout the evolutionary history of this family of enzymes[27]. It has been suggested that NAD$^+$·NADH is captured within the enzyme during its translation and passively involved in the folding process of the protein[26]. In this study, we decided to employ *ino1* of *M. tuberculosis*, since this gene is known to function efficiently in *M. tuberculosis*, and the enzyme it encodes has been well characterized, both biochemically and structurally[23]. Therefore, once the enzyme had been produced in *B. subtilis*, it was intended to function as MI1PS. However, despite the detection of specific MI1PS mRNA and protein in strain MC001 (Fig. 2), no specific activity was detected (Fig. 3a). However, when *pbuE* was inactivated in *B. subtilis*, MI1PS became active as a functional enzyme (Fig. 3a). *B. subtilis pbuE* encodes a purine base/nucleoside efflux pump, which is induced when intracellular concentrations of purine bases increase[36] and thus helps to maintain the homeostasis of purine bases/nucleosides in cells, although it remains unclear as to why intracellular levels of purine bases/nucleosides are kept low under normal conditions. The inactivation of *pbuE* might increase the intracellular pool of purine bases/nucleosides, which could then be used for the biosynthesis of NAD$^+$·NADH (Fig. 3b). Previously, the apparent $K_D$ value for NAD$^+$ binding to *M. tuberculosis* MI1PS was reported to be $36 \pm 4$ μM, which is much weaker than values reported for other, similar enzymes characterized from *Archaeoglobus fulgidus* (1 μM) and *Arabidopsis thaliana* (~0.2 μM)[25]. In addition, *B. subtilis* cell extracts, in which *M. tuberculosis* MI1PS encoded by *ino1Mt* was expressed with active *pbuE*, exhibited no specific activity *in vitro* even in the presence of 10 mM NAD$^+$ (Fig. 3a). It is likely that under natural conditions in *B. subtilis* the level of NAD$^+$·NADH is too low to make *M. tuberculosis* MI1PS functional or to stabilize it as an active enzyme.

It has previously been reported that the MI1PS reaction could be the rate-limiting step in natural *myo*-inositol biosynthesis[31]. In addition, we found that MI1PS became active in *B. subtilis* only when the intracellular level of NAD$^+$·NADH was elevated. Therefore, one of the key elements for improved *scyllo*-inositol production is the performance of MI1PS. It was only recently that an uncharacterized *ino1* was found in the genome of a strain of *Bacillus thuringiensis*[27]. Although nothing is known about its inositol biosynthesis, this bacterium is one of the closest relatives of *B. subtilis*, and this enzyme might therefore be more suitable in *B. subtilis* than the one from *M. tuberculosis*. On the other hand, it was reported that Mck1, a homolog of glycogen synthase kinase 3, could act as a novel positive regulator of *de novo myo*-inositol synthesis in *S. cerevisiae*[37], suggesting that there might be other such additional factors required to make the enzyme function properly.

In the version of a *B. subtilis* cell factory for the production of *scyllo*-inositol from glucose reported here, *scyllo*-inositol corresponding to 10% of the initial glucose was produced during the later period of an extended culturing process (Fig. 4). These results suggested that 90% of the glucose was used for energy and/ or as a carbon source for bacterial growth. In *B. subtilis*, the phosphoenolpyruvate-dependent phosphotransferase system is the major mechanism for the uptake of glucose, importing glucose as G6P, while another mechanism, involving the glucose transporter GlcP and glucose kinase GlcK, is thought to play a minor role[29,30]. Therefore, we tried to enhance the latter mechanism to feed more G6P by overexpressing both *glcP* and *glcK* simultaneously and constitutively (Fig. 1). Although the results indicated accelerated *scyllo*-inositol production, the final

concentration of scyllo-inositol did not substantially increase (Fig. 4). It is likely that the major metabolic pathways, such as glycolysis and the pentose phosphate pathway, are so efficient in G6P consumption that only a limited amount of G6P is available for conversion into scyllo-inositol. In order to redirect more G6P to scyllo-inositol production, we plan to manipulate the two key enzymes that metabolize G6P: Pgi, which converts G6P into fructose-6-phosphate in glycolysis, and Zwf, which transforms G6P into D-glucono-1,5-lactone-6P in the pentose phosphate pathway. However, since the manipulation of these two genes could possibly disturb the holistic metabolism of the cell system, we will need to first devise a proper strategy to avoid such problems.

To conclude, in this study we have demonstrated the first example of a bacterial cell factory for the production of scyllo-inositol from glucose, by coupling the process by which glucose is converted into myo-inositol with the previously established process of converting myo-inositol into scyllo-inositol. To enable the former process, pbuE had to first be inactivated to elevate the intracellular concentration of NAD$^+$·NADH and make MI1PS functional. At present, glucose is sold at the price about USD 0.45-0.60 per kg, while myo-inositol is at USD 10.00-30.00 per kg. The bacterial cell factory created in this study gave a production efficiency about $2 \, \text{g L}^{-1}$ of scyllo-inositol from $20 \, \text{g L}^{-1}$ of glucose, and the produced scyllo-inositol would not be cheaper than USD 4.50-6.00 per kg considering the material cost. On the other hand, one of the most efficient cell factories previously created was capable of producing $27.6 \, \text{g L}^{-1}$ of scyllo-inositol from $50 \, \text{g L}^{-1}$ of myo-inositol[15]. In this case, the produced scyllo-inositol would be more expensive than USD 18.12-54.35 per kg. Therefore, the new cell factory possibly makes the scyllo-inositol production at least 3-times cheaper than the previous ones, which could be improved in our future studies involving the strategies mentioned above. scyllo-inositol is an important compound that will help us to challenge the growing problem of Alzheimer's disease, and our bacterial cell factory will ensure an inexpensive way to produce this rare compound.

## Methods

**Bacterial strains, culture conditions, and primers.** Bacterial strains and oligo-nucleotide primers used in this study are listed in Supplementary Tables 1 and 2, respectively. Bacterial strains were maintained in lysogeny broth (LB) medium[38], which was supplemented with antibiotics, including $5 \, \mu\text{g mL}^{-1}$ chloramphenicol, $0.5 \, \mu\text{g mL}^{-1}$ erythromycin, $5 \, \mu\text{g mL}^{-1}$ kanamycin, $4 \, \mu\text{g mL}^{-1}$ phleomycin, and $100 \, \mu\text{g mL}^{-1}$ spectinomycin, as required. For the production of inositol, bacterial strains were grown at 37 °C with shaking at 180 rpm in Soytone medium, containing 4% (w/v) Bacto Soytone (Becton, Dickinson and Co., Franklin Lakes, NJ, USA), 0.5% (w/v) Bacto yeast extract (Becton, Dickinson and Co.), and $20 \, \text{g L}^{-1}$ (w/v) glucose.

**Construction of bacterial strains.** B. subtilis 168 was our standard strain. Strains MYI04, TM0310, and KU302 were constructed as previously described[15,18,39]. Strain YDHLd [pbuE::pMutin2(erm)] was obtained from the National Bioresource Project at the National Genetics Research Institute, Mishima, Japan.

Strain KS001 was constructed from MYI04 using the marker-free deletion technique, as follows[39] (Supplementary Fig. 3). Four PCR fragments, including fragments A, B, C, and the mazF cassette were prepared. Fragment A corresponded to upstream of the iolG region as the target of deletion, Fragment B downstream of the deletion target, Fragment C inside the deletion, and the mazF cassette contained mazF for a suicidal toxin under the control of an IPTG-inducible promoter (Pspac) regulated by lacI and a spectinomycin-resistance gene. Fragments A, B, and C were amplified from genomic DNA of strain 168 by PCR using the primer pairs DiolGAF/DiolGAR, DiolGBF/DiolGBR, and DiolGCF/DiolGCR (Supplementary Table 2), respectively, and the mazF cassette was amplified from DNA of TM0310 with MazFF/MazFR[39–41]. These four fragments, A, B, the mazF cassette, and C were ligated in this order by PCR using the primer pair DiolGAF/DiolGCR. The ligated PCR product was used to transform strain MYI04 into a spectinomycin-resistant pop-in mutant with integration of the PCR product through homologous recombination within the regions corresponding to fragments A and C. The spectinomycin-resistant transformants obtained in this way were grown in the absence of spectinomycin and then screened on IPTG-containing

plates for spectinomycin-sensitive mutants, which could appear following intrachromosomal recombination between the two direct-repeat regions corresponding to fragment B to pop-out the mazF cassette together with the target of deletion. One of the spectinomycin-sensitive mutants was selected to have the desired marker-free deletion of iolG, to yield strain KS001.

Strain TK001 was constructed as follows. Three PCR fragments, D, E, and F were prepared. Fragment D, containing the C-terminal region of the amyE locus, a chloramphenicol resistance gene, and the promoter of the rpsO gene (PrpsO), was amplified from strain 168 DNA by PCR using the primer pair AmyAF/AmyAB. Fragment E, containing the N-terminal region of the amyE locus, was amplified from DNA of strain 168 by PCR using the primer pair AmyBF/AmyBB. Fragment F, containing ino1 of M. tuberculosis (ino1Mt), codon-optimized from the original (https://mycobrowser.epfl.ch/genes/Rv0046c), was amplified from custom-made synthetic DNA (Eurofins Scientific, Brussels, Belgium) by PCR using the primer pair ino1F/ino1B to generate the ribosome-binding site of rpsO and the tufA terminator of B. subtilis, flanking the head and tail, respectively (Supplementary Fig. 1). The three fragments D, F, and E were ligated in this order by PCR using the primer pair AmyAF/AmyEBB. The ligated PCR product was used to transform strain 168, making it resistant to chloramphenicol, and resulting in strain TK001, whose correct construction was confirmed by DNA sequencing.

Strain TK002 was constructed as follows. Two PCR fragments, G and H, were prepared. Fragment G, containing the C-terminal region of the amyE locus, the chloramphenicol resistance gene, PrpsO, and the codon-optimized ino1 gene fused with a His-tag (His$_6$) in the C-terminus (ino1Mt-His$_6$), was amplified from TK001 by PCR using the primer pair AmyAF/inohis. Fragment H, containing the His-tag and the tufA terminator followed by the N-terminal region of amyE, was amplified using the primer pair hister/AmyBB from TK001 DNA. The two fragments, G and H, were ligated by PCR using the primer pairs AmyAF/AmyBB. The ligated PCR product was used to transform strain 168, making it resistant to chloramphenicol, and resulting in strain TK002, whose correct construction was confirmed by DNA sequencing.

Strain TK003 was constructed as follows. Three PCR fragments I, J, and K were prepared. Fragment I, containing the C-terminal part of amyE and the chloramphenicol resistance gene, was amplified from KU302 DNA using the primer pair AmyAF/AmyAB. Fragment J, containing the ribosome-binding site of rpsO, the ino1Mt-His$_6$ gene, the tufA terminator, and the N-terminal part of AmyE was amplified using the primer pair ino1F/AmyBB from TK002 DNA. Fragment K, which contained the promoter of ybfK from B. subtilis, was amplified from DNA of strain 168 using the primer pair PybfKF/PybfKB. The three fragments I, K, and J were ligated in this order by PCR using the primer pair AmyAF/AmyBB. The ligated PCR product was used to transform strain 168, making it resistant to chloramphenicol, and resulting in strain TK003, whose correct construction was confirmed by DNA sequencing.

Strain MC021 was constructed as follows. Three PCR fragments, L, M, and N were prepared. Fragment L, containing the C-terminal region of the pbuE locus, was amplified from DNA of strain 168 by PCR using the primer pair pbuEAF/pbuEAB. Fragment M, which contained the N-terminal region of the pbuE locus, was amplified from DNA of strain 168 using the primer pair pbuEBF/pbuEBB. Fragment N, containing the chloramphenicol resistance gene and the ino1Mt-His$_6$ gene, was amplified from TK002 using the primer pair cmino1F/cmino1B. The three fragments L, N, and M were ligated in this order by PCR using the primer pair pbuEAF/pbuEBB. The ligated PCR product was used to transform strain 168, making it resistant to chloramphenicol and resulting in strain MC021, whose correct construction was confirmed by DNA sequencing.

Strain MC030 was constructed as follows. Six PCR fragments, O, P, Q, R, S, and T were prepared. Fragment O, containing the C-terminal region of the epr locus, was amplified from DNA of strain 168 by PCR using the primer pair EprAF/EprAB. Fragment P, containing PrpsO, was amplified from TK002 using the primer pair PrpsOA/PrpsOB. Fragment Q, containing glcP, was amplified from DNA of strain 168 using the primer pair GlcPF/GlcPB. Fragment R, containing glcK, was amplified from DNA of strain 168 using the primer pair GlcKF/GlcKB. Fragment S, containing the phleomycin resistance gene (ble), was amplified from the cassette upp-phleo-cI[42] using the primer pair PhleoF/PhleoB. Fragment T, containing the N-terminal part of epr, was amplified from DNA of strain 168 using the primer pair EprBF/EprBB. The six fragments O, P, Q, R, S, and T were ligated by PCR using the primer pair EprAF/EprBB and used to transform strain 168, making it resistant to phleomycin and resulting in strain MC030, whose correct construction was confirmed by DNA sequencing.

The genetic elements constructed in the above mutant strains were combined in various combinations, as follows. Strain MC001 [ΔiolABCDEFGHIJ ΔiolX ΔiolR amyE::(PrpsO-ino1Mt-His$_6$ cat)] was made from KS001 transformed to be resistant to chloramphenicol by using TK002 DNA. MC010 [ΔiolABCDEFGHIJ ΔiolX ΔiolR pbuE::pMutin2(erm)] was made from KS001 transformed to be resistant to erythromycin by using YDHLd DNA. MC011 [ΔiolABCDEFGHIJ ΔiolX ΔiolR amyE::(PrpsO-ino1Mt-His$_6$ cat) pbuE::pMutin2(erm)] was made from MC001 transformed to be resistant to erythromycin by using YDHLd DNA. MC020 [ΔiolABCDEF ΔiolHIJ ΔiolX ΔiolR amyE::(PrpsO-iolG-iolW-iolT kan)] was made from KU302 transformed to be resistant to kanamycin by using plasmid pCm::Nm DNA[43]. MC022 [ΔiolABCDEF ΔiolHIJ ΔiolX ΔiolR amyE::(PrpsO-iolG-iolW-iolT kan) pbuE::(PrpsO-ino1Mt-His$_6$ cat)] was made from MC020 transformed to be resistant to chloramphenicol by using MC021 DNA. MC031 [ΔiolABCDEF,

Δ*iolHIJ* Δ*iolX* Δ*iolR* *amyE*::(P*rpsO*-*iolG*-*iolW*-*iolT kan*) *pbuE*::(P*rpsO*-*ino1Mt*-*His6 cat*) *epr*::(P*rpsO*-*glcP*-*glcK ble*)] was made from MC022 transformed to be resistant to phleomycin by using MC030 DNA.

**Northern blot analysis**. *B. subtilis* strains were grown in Soytone medium for 24 h at 37 °C with shaking. Total RNAs were extracted from the cells and purified as previously described[44]. The RNA samples were subjected to a Northern blot analysis using a DIG-labeled RNA probe specific for *M. tuberculosis ino1* as follows. A DNA fragment corresponding to part of the *ino1*-coding region of *M. tuberculosis* was PCR-amplified from strain TK002 using the primer pair DIGinoMtF/DIGinoMtB and with the introduction of a T7 RNA polymerase promoter sequence at the 3′-terminus. The PCR product was used as the template for *in vitro* transcription using a DIG RNA labeling kit (SP6/T7) (Roche Diagnostics, Basel, Switzerland) to produce the DIG-labeled RNA probe. The RNA samples were subjected to agarose gel electrophoresis, transferred to a positively charged nylon membrane (Roche Diagnostics), and hybridized with the DIG-labeled probe according to the manufacturer's instructions. The DIG-labeled RNA probe hybridized to *ino1* mRNA was detected using a DIG luminescence detection kit (Roche Diagnostics) and visualized with ChemiDoc (Bio-Rad, Hercules, CA, USA)[45].

**Western blot analysis**. *B. subtilis* strains were grown in Soytone medium at 37 °C with shaking. Bacterial cells were harvested when the cultures reached 50 units at $OD_{600}$ and washed three times with cold lysis buffer containing 20 mM Tris/HCl (pH 8), 10 mM NaCl, 10 mM EGTA, 5 mM EDTA, and 50 mM 2-mercaptoethanol, then stored at −80 °C. The cells were suspended in 10 mL lysis buffer containing 100 μL Halt Protease Inhibitor Single-Use Cocktail (Thermo Fisher Scientific, Waltham, MA, USA) and 35 μL 2-mercaptoethanol, then disrupted by three passages at 120 bar in Avestin Emulsiflex B15 cell disruptor (ATA Scientific, New South Wales, Australia). Following centrifugation, supernatants were mixed with 1.0% (w/w) protamine sulfate to precipitate nucleic acids. After further centrifugation, supernatants were subjected to 12% PAGE and the proteins separated in the gel were transferred to an iBlot PVDF membrane (Thermo Fisher Scientific, Waltham, MA, USA), where they were subsequently reacted with 2,000-times diluted THE$^{TM}$ His-tag antibodies (Genescript, Piscataway, NJ, USA) and thereafter with 20,000-times diluted secondary anti-mouse IgG antibodies (Sigma Aldrich, St. Louis, MO, USA). Proteins were revealed using Western BLoT Hyper HRP Substrate (Takara Bio, Shiga, Japan) and visualized with ChemiDoc (Bio-Rad).

**MI1PS assay**. Bacterial cells of 50 $OD_{600}$ units were harvested after 24 h of culturing in Soytone medium, washed twice, and suspended in 10 mL buffer containing 50 mM Tris-acetate (pH 7.4), 0.1 mM EDTA, and 1.0 mM ammonium acetate. The suspension was subjected to three passages at 120 bar in Avestin Emulsiflex B15 cell disruptor to prepare the cell extracts. MI1PS enzyme activity in the cell extracts was measured as previously described[46]. The reaction was initiated by mixing 0.2 mL of cell extract, 0.2 mL of 5 mM G6P, and 0.1 mL of 50 mM NAD$^+$; the mixture was then incubated at 37 °C for 10 min. The reaction was terminated by adding 0.1 mL of 20% trichloroacetic acid and the mixture was centrifuged to precipitate denatured proteins. Next, 0.5 mL of supernatant was mixed with 0.5 mL of 0.2 M NaIO$_4$ and incubated at 37 °C for 1 h to free phosphate from MI1P. The reaction was terminated by adding 1 mL of l M Na$_2$SO$_3$ to destroy any excess NaIO$_4$ and 2 mL of coloration reagent solution containing 600 mM H$_2$SO$_4$, 0.5% (w/v) ammonium molybdate, and 2% (w/v) ascorbic acid was added. The mixture was incubated at 37 °C for 1.5 h, and then absorbance at 660 and 820 nm was measured to determine the concentration of phosphate produced.

**NAD$^+$·NADH measurement**. *B. subtilis* strains were grown in Soytone medium at 37 °C with shaking. NAD$^+$·NADH levels were assessed using an EnzyChrom NAD$^+$·NADH Assay Kit (BioAssay Systems, Hayward, CA, USA), according to the manufacturer's instructions. The total intracellular concentration of NAD$^+$ plus NADH was calculated assuming that 1 $OD_{600}$ unit corresponded to $10^9$ cells and that each cell had a volume of 1.41 μm$^3$[47].

***scyllo*-Inositol measurement**. Aliquots of the bacterial culture media were passed through Amicon Ultra 0.5 mL 3K centrifugal filters (Millipore, Billerica, MA, USA) and subjected to high-performance liquid chromatography (HPLC) with a refractive index detector (LaChrom Elite: HITACHI High Technologies, Tokyo, Japan). HPLC was performed using a COSMOSIL Sugar-D column (4.6 × 250 mm) (Nacalai Tesque, Kyoto, Japan) maintained at 28 °C with a flow of acetonitrile/water (80/20) at 2 mL min$^{-1}$. The retention time was used to identify *scyllo*-inositol and refractive index units were used to calculate the concentration[15].

**Statistics and reproducibility**. All the experiments were repeated independently at least for three times, and the quantitative data were processed to calculate means ± SEM. Statistical significances were calculated using the Mann–Whitney U test with Prism (GraphPad software).

**Reporting summary**. Further information on research design is available in the Nature Research Reporting Summary linked to this article.

## Data availability
The datasets generated during and/or analyzed during the current study are available from the corresponding author on reasonable request.

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

## Acknowledgements
The authors thank the National Bioresource Project (NBRP) at the National Genetics Research Institute, Mishima, Japan, for providing strain YDHLd. This work was financially supported by the Ministry of Education, Culture, Sports, Science and Technology (MEXT), Japan, under Special Coordination Funds for Promoting Science and Technology, Creation of Innovative Centers for Advanced Interdisciplinary Research Areas, and KAKENHI 17K19237 and 18H02128 to K.Y.

## Author contributions
K.Y. conceived the study and wrote the final manuscript. C.M. designed and performed the experiments. C.K., S.K., and K.T. contributed to the construction of the bacterial strains. All authors were involved in drafting the manuscript, and S.I. contributed to the critical reading and revising of the manuscript.

## Competing interests
The authors declare no competing interests.
