## [Peer Review File · Communications Biology]

Reviewers' comments:

Reviewer #1 (Remarks to the Author):

The paper "A bacterial cell factory converting glucose into scyllo-inositol, a therapeutic agent for Alzheimer's disease" by Michon et al. describes the use of the Gram-positive bacterium *B. subtilis* as a factory for the production of a relevant drug. The results are interesting, however I question the novelty of this approach, even with respect to previous work published by the same authors.

Specific comments:

Line 19.- The abstract needs to be rewritten to clarify what the authors did, why and how.

Line 23-25.- Confusing sentence

Line 34- 35.- Add relevant citations

Line 37-39 .- Rewrite sentence

Line 41- 44.- More appropriate in the discussion section

Line 51.- which enzyme, why is it expensive?

Figure 1.- I encourage the authors to improve the Figure 1

Line 60.- "enables a *B. subtilis* cell factory to" Rewrite, delete cell factory. The properties mentioned are prior to the engineering of *B. subtilis*

Line 62-63. "One of the cell factories to be most recently created was capable of producing 27.6 g L" Rewrite

Line 66-67.- Clarify. Why is the product cheaper than the starting material? MI is very expensive? what is the source of MI?

Line 75.- Although mycobacterium is classified as Gram-positive it possesses unique lipids that deeply impact its properties.

Line 78-80.- Again, not entirely clear to me why this approach is superior/substantially different to others previously published by the same group (e.g., *Microb Cell Fact.* 2011 Sep 7;10:69. doi: 10.1186/1475-2859-10-69).

Figure 2.- Show molecular weight in Figure 2 and load controls

Line 94-98.- The lack of activity could be related to the addition of his tag. Furthermore, these experiments do not reveal whether the protein is misfolded or not properly translated. Additional structural experiments using e.g., circular dichroism should be performed. MI1P should be detected with specific antibodies to check its proper production.

Line 104-105.- Is this a new result or from the cited article? Clarify

Line 114.- The levels are higher in YDHLd, the parental strain, but in MC011 no measurements are shown. The NAD NADH levels in the engineered strains need to be quantified.

Line 157.- "since this gene is known to function efficiently in *B. subtilis* ," Which article describes this? The first result shows that *ino1* from *M. tuberculosis* doesn't work in *B. subtilis*

Line 159-160.- "are Gram-positive bacteria, so we assumed that the *M. tuberculosis* enzyme might also function in *B. subtilis*" Review or delete this sentence. The membrane of *Mycobacterium* is very complex and its GC %, growth rate and growth requirements are very different from those of *B. subtilis*.

Line 162-163.- "Therefore, we assumed that the enzyme produced in *B. subtilis* could have been misfolded and therefore inactive." Review, see previous note about this affirmation

Line 173-174.- Data presented indicate 10mM NAD NADH. Discuss.

Line 178-179.- "It was only recently that an uncharacterized *ino1* present in the genome of a strain of *Bacillus thuringiensis* was found." Rewrite sentence

Line 180-181.- "the closest relatives of *B. subtilis*, and this enzyme might therefore be more suitable in *B. subtilis* than the one from *M. tuberculosis*." Why was the one from *M. tuberculosis* not chosen? Clarify please

Line 181.- "Mck1" From which strain?

Line 190-193.- Was the SI production measured per cell instead of per liter? Maybe bacteria are growing faster and therefore the production of SI is "accelerated".

Line 197-198.- How do you plan to solve this?

Line 302.- Add loading controls as 16S, or at least other housekeeping gene/s.

Line 316.- Add loading controls

Line 332.- "50 OD600 units" Why did you assume this? Cite previous CFU counts or references for to support this

Line 336.- "0.2 mL cell extract" Why did you assume this? Cite previous CFU counts or references to support this

Line 349.- Why did you assume this? Cite previous CFU counts or previous work

Reviewer #2 (Remarks to the Author):

The paper by Michon et al describes the design of a recombinant *Bacillus subtilis* that can make the valuable drug synthon scyllo-inositol (SI) from glucose. The work builds on previous research to biosynthetically produce SI, with the novelty that the starting material is glucose rather than more expensive myo-inositol. Overall, I found the paper interesting, well-written and a first step to producing SI from an inexpensive substrate.

Minor comments

1. Line 41 should be '..fewer amyloid plaques.'
2. Line 42, state the sample size
3. Line 92 replace 'barely' with 'not'
4. Fig 1. Would it be possible to include some colour here to highlight the newly-introduced genes/enzymes?

Reviewer #3 (Remarks to the Author):

MS title: A bacterial cell factory converting glucose into scyllo-inositol, a therapeutic agent for Alzheimer's disease

Summary: The authors introduce a gene (*ino1*) encoding myo-inositol-1-phosphate synthase (MI1PS) from *Mycobacterium tuberculosis* to *Bacillus subtilis* strain 168. Transcript and protein were produced using different promoters, but activity was not detected. The authors found that a *Bacillus subtilis* strain with an inactivated *pbuE* gene displayed an increase in intracellular NAD/NADH concentrations. *pbuE* inactivation, disruption of *iolABCDEFGHIJ/iolX/iolR* and introduction of *ino1* resulted in a strain with substantial levels of MI1PS activity. The artificial pathway comprising *iolG/iolW/iolT* was added to this strain, generating a strain with the capability of producing scyllo-inositol from glucose. Further addition of *GlcP* (glucose transporter) and *GlcK* (glucose kinase) resulted in the enhancement of scyllo-inositol production rates.

General comments: The generation of a *Bacillus* strain that produces gram levels of scyllo-inositol is impressive, and the results are novel. The reviewer was not aware of the value of scyllo-inositol, but its activity towards dementia may be useful in the future. As there is no similar alternative for scyllo-inositol production, the study should provide important information to those engaged in the field. The experiments have been performed carefully, and the manuscript is well written and was relatively easy to follow. There are some points however that might be considered. They are not directly related to the overall conclusions, but consideration would help clarify some portions of the manuscript.

1. Huang and Hernick reported that the MI1PS from *Mycobacterium tuberculosis* folded improperly in

Escherichia coli (lacking an important alpha helix), but was functionally produced in *Mycobacterium smegmatis*. It was slightly surprising that the authors chose this particular protein for constructing the cell factory (in *Bacillus*). Is it correct that there were not so many other alternatives, such as homologs from *Bacillus* and the recently identified homolog from *Bacillus thuringiensis*? The fact that both organisms are Gram-positive is not such a strong common denominator.

2. Regarding the isolation or examination of *Bacillus subtilis* YDHLd, if possible, some points should be clarified. Why was this strain examined in the first place? Zakataeva et al report that the PbuE pump functions not only on purine bases, but also purine nucleosides. Did the authors presume that it might also function on NAD export (or the export of its precursors in biosynthesis)? The reviewer has no problem with the coincidental discovery of this strain showing a significant elevation in NAD/NADH concentrations, but why did the authors examine this strain?

3. Do the authors have a [pbuE::pMutin2(erm) amyE::(PrpsO-ino1Mt-His6 cat)] strain? If so, this would help clarify what triggers functional expression of the *ino1* gene.

Other points:

It might be convenient for the reader if the authors presented the entire iol catabolism/metabolism in a supplementary figure.

The reviewer is not comfortable in the description and evaluation of "yield" (eg lines 142, 192) It is hard to determine the yield, as the concentration of scyllo-inositol is still increasing at 100 h.

Particularly, production rates are still steady for strain MC031. This should be considered or rephrased.

Line 38: where it covers amyloid β -proteins, the term covers is vague and could be replaced.

Line 46: delete the

Line 50: MI and then

Line 55: why does iolX "ensure" catabolism?

Line 60: "cell factory" with quotation marks should appear here?

Line 63: The starting material should be indicated in this sentence.

Line 155: Add period

Line 193: might be so efficient in G6P consumption that the limited amount of G6P available for conversion into SI. (sentence needs to be rephrased)

Re: manuscript No. COMMSBIO-19-1206A

Please find our point-by-point responses to the comments from the three referees as follows:

Referee #1: Microbiome engineering for therapeutic purposes

The paper “A bacterial cell factory converting glucose into scyllo-inositol, a therapeutic agent for Alzheimer’s disease” by Michon et al. describes the use of the Gram-positive bacterium *B. subtilis* as a factory for the production of a relevant drug. The results are interesting, however I question the novelty of this approach, even with respect to previous work published by the same authors.

Authors’ response (AU): First of all, we so much appreciate the serious and critical reading made by the reviewer. As stated in the main text, we have demonstrated the “first” example of a bacterial cell factory for the production of SI from glucose, not from MI as we reported previously. Therefore, the novelty of this study is obvious. SI is an important compound that helps us to challenge the growing problem of Alzheimer’s disease, and our bacterial cell factory will ensure an inexpensive way to produce this rare compound. Please find more explanation below.

Specific comments:

Line 19.- The abstract needs to be rewritten to clarify what the authors did, why and how.

AU: We and the referee have different views, and we believe the abstract clearly describes why and how we did.

Line 23-25.- Confusing sentence

AU: The long sentence was revised splitting into two sentences (L. 22-25).

Line 34- 35.- Add relevant citations

AU: A reference is added (L.35).

Line 37-39 .- Rewrite sentence

AU: The sentence was rewritten (L. 35-39).

Line 41- 44.- More appropriate in the discussion section

AU: We and the referee have different views, and we believe the facts are informative for readers as part of introduction to learn how SI has been studied for its effectiveness to Alzheimer’s disease.

Line 51.- which enzyme, why is it expensive?

AU: The conversion of phytic acid into SI requires phytase and inositol dehydrogenases. And the in vitro reactions require addition of NAD^+ and NADPH as the essential cofactors. The enzyme preparation and the cofactors make the process expensive.

Figure 1.- I encourage the authors to improve the Figure 1

AU: The Fig. 1 was improved to show some of the enzymes in color to distinguish the one introduced heterogeneously and the others over-expressed.

Line 60.- “enables a *B. subtilis* cell factory to” Rewrite, delete cell factory. The properties mentioned are prior to the engineering of *B. subtilis*

AU: The sentence was revised as suggested (L. 59-61).

Line 62-63. “One of the cell factories to be most recently created was capable of producing 27.6 g L⁻¹” Rewrite

AU: The sentence was revised (L.62-63).

Line 66-67.- Clarify. Why is the product cheaper than the starting material? MI is very expensive? what is the source of MI?

AU: The bioconversion is efficient enough to enable a high productivity, but the SI produced can never be cheaper than the starting material MI. (L. 66-67)

Line 75.- Although mycobacterium is classified as Gram⁺positive it possesses unique lipids that deeply impact its properties.

AU: Indeed, *Mycobacterium tuberculosis* has specific and essential phospholipids containing myo-inositol, and thus the MI1PS enzyme is indispensable for this bacterium.

Line 78-80.- Again, not entirely clear to me why this approach is superior/substantially different to others previously published by the same group (e.g., *Microb Cell Fact.* 2011 Sep 7;10:69. doi: 10.1186/1475-2859-10-69).

AU: As mentioned above, in this study we demonstrated the “first” example of a bacterial cell factory for the production of SI from glucose by coupling the process by which glucose is converted into MI with the previously established process of converting MI into SI. To enable the former process, *pbuE* had to first be inactivated to elevate the intracellular concentration of NAD⁺·NADH and make MI1PS functional. At present, glucose is sold at the price about USD 0.45-0.60 per kg, while MI is at USD 10.00-30.00 per kg. The bacterial cell factory created in this study gave the production efficiency about 2 g L⁻¹ of SI from 20 g L⁻¹ of glucose, and the produced SI would not be cheaper than USD 4.50-6.00 per kg considering the material cost. On the other hand, one of the most efficient cell factories previously created was capable of producing 27.6 g L⁻¹ of SI from 50 g L⁻¹ of MI 14. In this case, the produced SI would be more expensive than USD 18.12-54.35 per kg. Therefore, the new cell factory possibly makes the SI production at least 3-times cheaper than the previous ones, which could be further improved in our future studies (L. 198-208). SI is an important compound that will help us to challenge the growing problem of Alzheimer’s disease, and our bacterial cell factory will ensure an inexpensive way to produce this rare compound.

Figure 2.- Show molecular weight in Figure 2 and load controls

AU: For the Northern and Western analyses (Fig. 2), the sizes of transcript and protein had been included in the figure legend already in the previous version. In this revision, we added the explanation that 30 micro-g of the RNA extracts and 50 micro-g of the proteins were loaded in each of the lanes, respectively (L. 484 and 486). The purpose of these experiments was to show the existence of the specific transcript and His-tagged protein, and it does not matter how much they were. Therefore, we do not require the loading controls. In any case, we demonstrated the enzyme activity later, which is more convincing to tell its expression.

Line 94-98.- The lack of activity could be related to the addition of his tag. Furthermore, these experiments do not reveal whether the protein is misfolded or not properly translated. Additional structural experiments using e.g., circular dichroism should be performed. MIIP should be detected with specific antibodies to check its proper production.

AU: The His-tag was not involved in the activity, since the protein turned out to be active later as shown in this study. There is no antibody that recognizes MIIP, and the requested experiment is impossible to perform.

Line 104-105.- Is this a new result or from the cited article? Clarify

AU: It was not shown previously that inactivation of *pbuE* resulted in the increased levels of NAD⁺ and NADH.

Line 114.- The levels are higher in YDHLd, the parental strain, but in MC011 no measurements are shown. The NAD NADH levels in the engineered strains need to be quantified.

AU: YDHLd and MC011 are genetically identical except for the introduced *ino1* and the enhanced *iol* genes, which have nothing to do with the total of NAD⁺ and NADH in the cell.

Line 157.- “since this gene is known to function efficiently in *B. subtilis*,” Which article describes this? The first result shows that *ino1* from *M. tuberculosis* doesn’t work in *B. subtilis*

AU: It was our mistake while editing the manuscript. This gene is known to function efficiently in *M. tuberculosis* (L. 158).

Line 159-160.- “are Gram-positive bacteria, so we assumed that the *M. tuberculosis* enzyme might also function in *B. subtilis*” Review or delete this sentence. The membrane of *Mycobacterium* is very complex and its GC %, growth rate and growth requirements are very different from those of *B. subtilis*.

AU: The sentence was deleted as suggested.

Line 162-163.- “Therefore, we assumed that the enzyme produced in *B. subtilis* could have been misfolded and therefore inactive.” Review, see previous note about this affirmation

AU: The sentence was also deleted.

Line 173-174.- Data presented indicate 10mM NAD NADH. Discuss.

AU: The in vitro enzyme assay was done in the presence of 10 mM NAD⁺, and MIIPS produced in *B. subtilis* was not active (L. 92). It was our mistake to have written the concentration as 1 mM and it should be 10 mM, since the stock solution of NAD⁺ was not 5 mM but 50 mM (L. 341).

Line 178-179.- "It was only recently that an uncharacterized *ino1* present in the genome of a strain of *Bacillus thuringiensis* was found." Rewrite sentence

AU: The sentence was revised appropriately (L. 177-178).

Line 180-181.- "the closest relatives of *B. subtilis*, and this enzyme might therefore be more suitable in *B. subtilis* than the one from *M. tuberculosis*." Why was the one from *M. tuberculosis* not chosen? Clarify please

AU: The *ino1* is known functional in *M. tuberculosis*, but the others have not been functionally identified yet. Therefore, we chose it for this study.

Line 181.- "Mck1" From which strain?

AU: This was from in *S. cerevisiae* (reference 37).

Line 190-193.- Was the SI production measured per cell instead of per liter? Maybe bacteria are growing faster and therefore the production of SI is "accelerated".

AU: The overexpression of *glcP* and *glcK* did not significantly affect the growth (Supplementary Fig. 2). (L. 138-139)

Line 197-198.- How do you plan to solve this?

AU: We plan to knock down these genes conditionally during the stationary phase when the SI production was mainly seen in this study. For the knocking down, we may consider the strategy to express artificially-designed specific antisense RNAs to inhibit translation of the targets.

Line 302.- Add loading controls as 16S, or at least other housekeeping gene/s.

Line 316.- Add loading controls

AU: As already explained above, for the Northern and Western analyses (Fig. 2), 30 micro-g of the RNA extracts and 50 micro-g of the proteins were loaded in each of the lanes, respectively (L. 484 and 486). The purpose of these experiments was to show the existence of the specific transcript and His-tagged protein, and it does not matter how much they were. Therefore, we do not require the loading controls.

Line 332.- “50 OD600 units” Why did you assume this? Cite previous CFU counts or references for to support this

Line 336.- “0.2 mL cell extract” Why did you assume this? Cite previous CFU counts or references to support this

Line 349.- Why did you assume this? Cite previous CFU counts or previous work

AU: Generally speaking, we can quantify the amount of cells in OD600 units. 50 OD600 units correspond to the cells contained in 50 ml of a culture at OD600=1.0. We disrupted the cells to obtain protein extracts as written in the text and used 0.2 ml of it for the analysis. 1 OD600 unit corresponded to 10^9 cells and that each cell had a volume of $1.41 \mu\text{m}^3$ (additional reference 46 was provided).

Referee #2: Industrial microbiology

The paper by Michon et al describes the design of a recombinant *Bacillus subtilis* that can make the valuable drug synthon scyllo-inositol (SI) from glucose. The work builds on previous research to biosynthetically produce SI, with the novelty that the starting material is glucose rather than more expensive myo-inositol. Overall, I found the paper interesting, well-written and a first step to producing SI from an inexpensive substrate.

AU: Thank you so much for the general comment that our paper is interesting and well-written. Please find our answers to your comments as follows:

Minor comments

1. Line 41 should be ‘..fewer amyloid plaques..’

AU: Corrected as suggested (L. 40).

2. Line 42, state the sample size

AU: The sample size was stated (L. 42).

3. Line 92 replace 'barely' with 'not'

AU: Corrected as suggested (L. 92).

4. Fig 1. Would it be possible to include some colour here to highlight the newly-introduced genes/enzymes?

AU: The Fig. 1 was improved to show some of the enzymes in color to distinguish the one introduced heterogeneously and the others over-expressed.

Referee #3: Biological chemistry, metabolic engineering of microbes

MS title: A bacterial cell factory converting glucose into scyllo-inositol, a therapeutic agent for Alzheimer’s disease

Summary: The authors introduce a gene (*ino1*) encoding myo-inositol-1-phosphate synthase (MI1PS) from

Mycobacterium tuberculosis to *Bacillus subtilis* strain 168. Transcript and protein were produced using different promoters, but activity was not detected. The authors found that a *Bacillus subtilis* strain with an inactivated *pbuE* gene displayed an increase in intracellular NAD/NADH concentrations. *pbuE* inactivation, disruption of *iolABCDEFGHIJ/iolX/iolR* and introduction of *ino1* resulted in a strain with substantial levels of MIIPS activity. The artificial pathway comprising *iolG/iolW/iolT* was added to this strain, generating a strain with the capability of producing scyllo-inositol from glucose. Further addition of *GlcP* (glucose transporter) and *GlcK* (glucose kinase) resulted in the enhancement of scyllo-inositol production rates.

General comments: The generation of a *Bacillus* strain that produces gram levels of scyllo-inositol is impressive, and the results are novel. The reviewer was not aware of the value of scyllo-inositol, but its activity towards dementia may be useful in the future. As there is no similar alternative for scyllo-inositol production, the study should provide important information to those engaged in the field. The experiments have been performed carefully, and the manuscript is well written and was relatively easy to follow. There are some points however that might be considered. They are not directly related to the overall conclusions, but consideration would help clarify some portions of the manuscript.

AU: We are glad that you found the novelty in our study, the experiments performed carefully, and the manuscript well written. Please find our responses to your comments as follows:

1. Huang and Hernick reported that the MIIPS from *Mycobacterium tuberculosis* folded improperly in *Escherichia coli* (lacking an important alpha helix), but was functionally produced in *Mycobacterium smegmatis*. It was slightly surprising that the authors chose this particular protein for constructing the cell factory (in *Bacillus*). Is it correct that there were not so many other alternatives, such as homologs from *Bacillus* and the recently identified homolog from *Bacillus thuringiensis*? The fact that both organisms are Gram-positive is not such a strong common denominator.

AU: We deleted the misleading sentences.

2. Regarding the isolation or examination of *Bacillus subtilis* YDHLd, if possible, some points should be clarified. Why was this strain examined in the first place? Zakataeva et al report that the *PbuE* pump functions not only on purine bases, but also purine nucleosides. Did the authors presume that it might also function on NAD export (or the export of its precursors in biosynthesis)? The reviewer has no problem with the coincidental discovery of this strain showing a significant elevation in NAD/NADH concentrations, but why did the authors examine this strain?

AU: It is known that *B. subtilis* secretes purine bases and nucleosides, when their concentrations get higher, although the physiological reason remains unknown yet. In this study, we found the total concentration of NAD⁺ and NADH was lower than that in *E.coli*, and speculated if the secretion of purine bases and nucleosides might be the reason. Therefore, we tested the inactivation of *pbuE* involved in the secretion to find the elevated levels of NAD⁺ and NADH. To be honest, our study included such a moment of serendipity

3. Do the authors have a [*pbuE::pMutin2(erm) amyE::(PrpsO-ino1Mt-His6 cat)*] strain? If so, this would help clarify

what triggers functional expression of the *ino1* gene.

AU: We have not done this, since the purpose of this study was to produce SI.

Other points:

It might be convenient for the reader if the authors presented the entire iol catabolism/metabolism in a supplementary figure.

AU: A more detailed explanation of the entire iol catabolism and its regulation in *B. subtilis* would not be required for the readers to understand the main story described in this study. We believe the references 10-14 can help the readers to learn further if needed.

The reviewer is not comfortable in the description and evaluation of "yield" (eg lines 142, 192) It is hard to determine the yield, as the concentration of scyllo-inositol is still increasing at 100 h. Particularly, production rates are still steady for strain MC031. This should be considered or rephrased.

AU: The word "yield" is replaced to "concentration" (L. 143 and 191) or "productivity" (L. 67).

Line 38: where it covers amyloid β -proteins, the term covers is vague and could be replaced.

AU: The sentence was revised (L. 38-39).

Line 46: delete the

AU: It was deleted. (L. 46)

Line 50: MI and then

AU: "and" was added. (L. 50)

Line 55: why does iolX "ensure" catabolism?

AU: Changed to "enables" (L.55).

Line 60: "cell factory" with quotation marks should appear here?

AU: The sentence was rewritten (L. 59-61).

Line 63: The starting material should be indicated in this sentence.

AU: 50 g L⁻¹ MI (L. 63).

Line 155: Add period

AU: It was added as suggested (L. 156).

Line 193: might be so efficient in G6P consumption that the limited amount of G6P available for conversion into SI.
(sentence needs to be rephrased)

AU: The sentence was rephrased (L. 191-193).

REVIEWERS' COMMENTS:

Reviewer #1 (Remarks to the Author):

The authors have addressed all my comments

Reviewer #3 (Remarks to the Author):

The previous comments by this reviewer have been sufficiently addressed. Looking at the revised manuscript, there are a few minor points for consideration.

L37/has shown promising should read has shown promise or has been shown to be promising

L38/becoming to? perhaps developing into or assembling into or forming

L177/It was only recently that or Only recently was an uncharacterized ino1 found

L192/ consumption that only a limited amount

L203/ gave a production efficiency of about

Responses to the reviewers' comments:

Reviewer #1 (Remarks to the Author):

The authors have addressed all my comments
>N/A

Reviewer #3 (Remarks to the Author):

The previous comments by this reviewer have been sufficiently addressed. Looking at the revised manuscript, there are a few minor points for consideration.

L37/has shown promising should read has shown promise or has been shown to be promising
>Revised as suggested.

L38/becoming to? perhaps developing into or assembling into or forming
>Revised as suggested.

L177/It was only recently that or Only recently was an uncharacterized ino1 found
>Revised as suggested.

L192/ consumption that only a limited amount
>Revised as suggested.

L203/ gave a production efficiency of about
>Revised as suggested.